# SCALING AUTONOMOUS DRIVING SAFETY WITH SYNTHETIC DATA

## ABSTRACT

Modern data-driven driving planners tend to perform suboptimally under safety-challenging cases that are underrepresented in training data. Given the difficulty in collecting real-world data to cover all possible corner cases, scaling synthetic training data to enhance planning safety is of considerable value. We propose SafeScale, a geometry-based generative driving simulation method that enables scalable generation of diverse driving corner cases. We compose novel scenes by combining visual and behavioral assets from real-world data, enabling precise scenario customization and ensuring synthetic data diversity. We employ a generative model to synthesize photorealistic camera observations along the simulated ego trajectory in novel scenes. We analyze the types of corner cases that the state-of-the-art planner struggles to handle and use SafeScale to synthesize corresponding scenarios as supplementary training data. Experiments on the NAVSIM dataset demonstrate that scaling up the amount of synthetic training data continuously improves the planner's performance on real-world data, exhibiting a clear data scaling effect. With up to 100K additional synthesized training scenarios, the state-of-the-art end-to-end planner achieves a 28.6% reduction in collision failure cases, a 34.6% reduction in near-collision failure cases, and a 20.9% reduction in driveable area deviation failure cases on the NAVSIM test set. Experiment results further show that synthetic data targeting each specific type of corner case yields highly selective improvements in planner performance under the responding scenario, and that the effects of synthetic data for different corner scenarios are independent and additive. To our knowledge, this work presents the first effective demonstration of improving real-world driving performance via synthetic data.

## 1 INTRODUCTION

Robustness in uncommon or dangerous driving scenarios is critical to the safety of autonomous driving systems. However, modern data-driven driving planning methods often exhibit sub-optimal performance when encountering driving situations that are infrequent or absent in the training data, commonly referred to as corner cases. Collecting large-scale real-world data for rare or dangerous driving scenarios is limited by efficiency, safety, and legality, particularly for scenarios that pose genuine risks of personal injury or property damage. Consequently, recent attention has turned to using scene reconstruction methods or generative models to simulate challenging driving scenarios in virtual environments as a substitute for real data. Yet, prior methods(Zhu et al., 2025; Xu et al., 2025) have primarily focused on using synthetic data as supplement test sets to evaluate planner performance in challenging situations. The use of large-scale synthetic data to enhance planning performance remains unexplored.

Simulating driving scenarios on demand in virtual environments requires simulation methods capable of precisely customizing 3D scenes, accurately controlling the ego vehicle's motion within the scenario, and ultimately generating realistic simulated sensor data. Early driving simulation methods based on virtual engines generate sensor data with significant gaps from real-world data. Simulation methods based on 3D reconstruction are limited by the restricted perspectives of original observations, making it difficult to simulate camera imaging under arbitrary ego trajectories. Large-scale 3D reconstruction of driving data also demands substantial computational resources. Existing generative driving video generation methods (i.e., driving world models) struggle to achieve centimeter-level control on scene content, such as the appearance details of traffic participants. Nor can they sim-

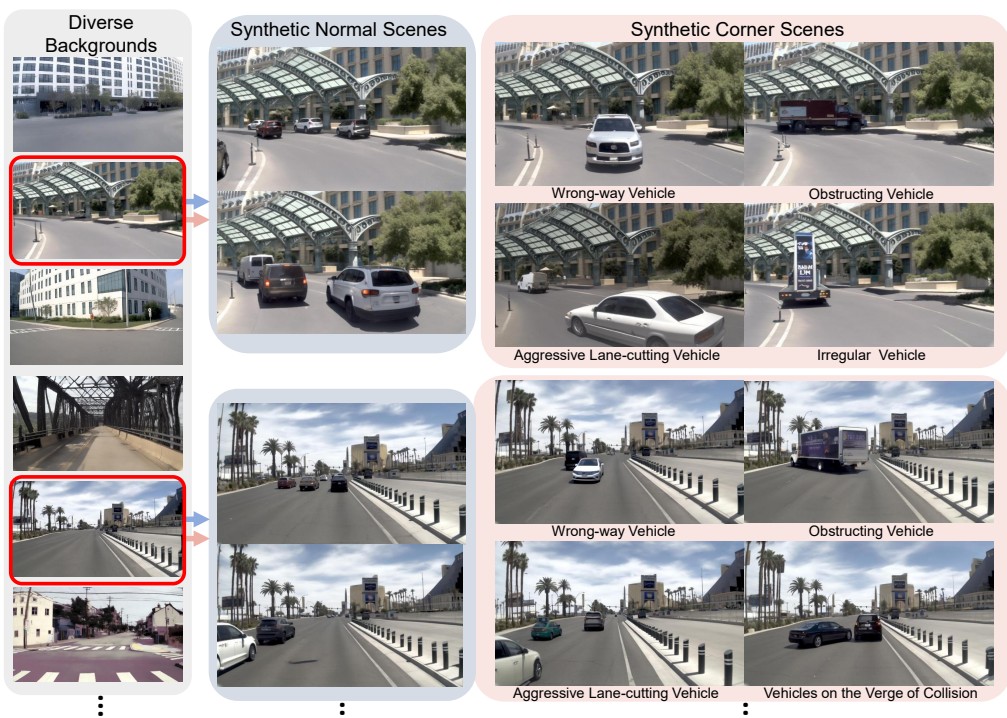

Figure 1: **Visualization of driving scenarios synthesized with SafeScale.** In any driving scene background, we can either randomly simulate normal driving scenes or, by precisely specifying the appearance, state, and behavior of any traffic participants, simulate diverse corner driving scenarios such as the presence of wrong-way vehicles, aggressive lane-cutting vehicles, unusual vehicles, and other participants on the verge of collision.

ulate ego vehicle trajectories with no execution errors. Due to these limitations, existing driving simulation methods are unable to generate targeted driving corner cases in a scalable manner. To address this, we propose SafeScale, a driving simulation method capable of accurately customizing 3D scenes and generating diverse driving corner cases at scale.

SafeScale is a geometry-based generative driving simulator. Leveraging rich driving scene assets extracted from real data, SafeScale performs explicit geometric modeling of 3D scenes, enabling precise driving scenario customization and scalable generation of diverse scenarios. We decompose large-scale real-world driving scenes into scene component assets, including static backgrounds, traffic participant appearances, and traffic participant behaviors. The visual assets are stored as colored 3D point clouds, which can be obtained at a low cost from data clips and can be edited efficiently. Based on specific data requirements, we randomly combine driving scene components from different real-world segments to generate unlimited, diverse new scenes. The explicit construction of 3D scenes allows us to precisely customize the category, appearance, and behavior of each traffic participant in the driving scenario, enabling batch generation of corner case driving data containing rare traffic participants or abnormal behaviors within diverse static driving backgrounds. Geometrically modeling driving scenes also enables us to objectively simulate the dynamic evolution of 3D scenes while strictly ensuring the long-term consistency of the scene contents. To simulate photorealistic views based on point cloud priors, SafeScale utilizes a generative model to synthesize camera observations along the simulated ego vehicle trajectories in the constructed scenes. Following the design of generative novel view synthesis methods(Wang et al., 2025), we can simulate ego vehicle trajectories within 3D scenes without execution error.

Experiments on the NAVSIM(Dauner et al., 2024) dataset demonstrate that driving data synthesized by SafeScale , when used as supplementary training data, can enhance the performance of end-to-end driving planner in real driving scenes. The use of synthetic data for training end-to-end planner exhibits a data scaling effect: scaling up the amount of synthetic training data is observed to be

able to continuously improve planner performance. Moreover, experiments show that corner driving scenarios simulated by SafeScale in a targeted and scalable manner can specifically improve model performance in corresponding corner cases, such as scenarios with rear-end collision risks. On the NAVSIM benchmark, training with SafeScale-generated data can reduce the number of failure cases of the current state-of-the-art end-to-end planning method DiffusionDrive(Liao et al., 2025) by up to 28% and achieve a +1.4% PDMS improvement.

Our contributions in this work are threefold:

- We propose SafeScale, a geometry-based generative driving simulation method that can synthesize diverse driving data in a scalable and precisely controllable manner.
- We empirically validate the data scaling effect of synthetic data on end-to-end planner training, and that synthetic corner-case data can specifically enhance the handling of corresponding corner cases.
- To our knowledge, this is the first work to successfully enhance real-world driving performance using synthetic data.

## 2 RELATED WORKS

**Driving simulation with graphics-based simulator or scene reconstruction.** Some recent synthetic benchmarks(Li et al., 2024a; Jia et al., 2024) in autonomous driving collect synthetic data with traditional graphics-based driving simulators like CARLA(Dosovitskiy et al., 2017). However, sensor data generated by traditional simulators often exhibit a substantial gap from real-world observations, which limits the applicability of synthetic data. Some methods utilize Neural Radiance Fields (NeRF)(Mildenhall et al., 2020) or 3D Gaussian Splatting (3DGS)(Kerbl et al., 2023) to perform 3D driving scene reconstruction for novel-view synthesis and scene editing(Chen et al., 2023; Zhou et al., 2024; Gao et al., 2025; Yan et al., 2024). However, their inability to supplement missing 3D scene information severely restricts their capability of viewpoint simulation and scene editing.

**Generative driving simulation.** In recent years, a growing number of studies have focused on driving video generation, often referred to as driving world models(Gao et al., 2024; Hu et al., 2023a; Wang et al., 2024a;b;c; Hu et al., 2022; Zhang et al., 2023; Chen et al., 2024). Very recently, some works have further focused on specific scene customization and precise viewpoint control to better meet the demands of driving simulation. ReSim(Yang et al., 2025a) trains generative models with heterogeneous Data from real-world and driving simulators to enhance the action controllability of video generation. X-Scene(Yang et al., 2025b) focuses on large-scale scene generation with spatial expansion while ensuring spatial consistency. Genesis(Guo et al., 2025) uses structured semantics to guide the consistent generation of multi-view driving videos and LiDAR data. DrivingSphere(Yan et al., 2025a) builds a closed-loop simulator with a generative model and the 4D occupancy representations of driving scenes. FreeVS(Wang et al., 2025) and StreetCrafter(Yan et al., 2025b) simulate camera views on novel driving trajectories in real 3D scenes through diffusion models. Challenger(Xu et al., 2025) modifies the driving behavior of one or multiple background vehicles in real scenes to generate adversarial driving videos as challenging test sets. DriveEditor(Liang et al., 2025) and SceneCrafter(Zhu et al., 2025) focus on real scene editing, including the insertion and removal of driving agents. While some previous works(Huang et al., 2025; Wang et al., 2024a) have validated the effect of synthetic training data on 3D object perception, the use of synthetic data to enhance end-to-end driving planning, particularly in corner cases, remains unexplored. In this work, we propose a scalable generative driving simulation method that enables precise customization of 3D scenes and demonstrate the effectiveness of using generated data as training input for improving the end-to-end driving planner.

## 3 METHOD

Safety-challenging synthetic data scaling aims to cover various unconventional driving scenarios in the real world, such as corner cases involving wrong-way vehicles, obstructing vehicles, and vehicles with irregular appearance, as illustrated in Fig. 1. To achieve this, both geometrically accurate scene customization and photorealistic camera view simulation are required, which impose high demands on the data synthesis pipeline. To enable controllable, diverse, and high-fidelity driving

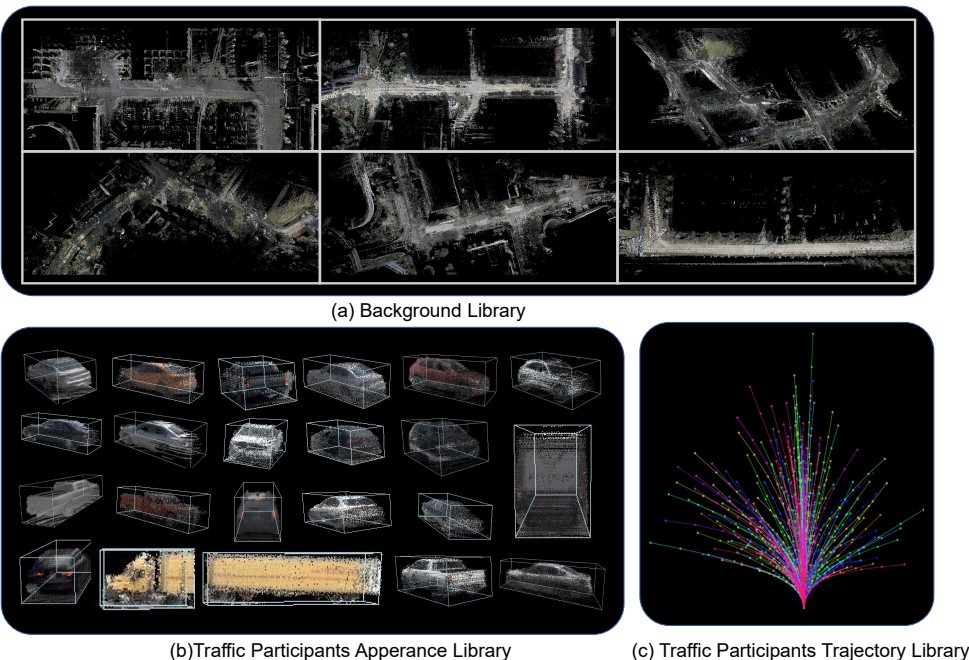

(a) Background Library

(b)Traffic Participants Apperance Library

(c) Traffic Participants Trajectory Library

Figure 2: **Illustration of driving scene element assets collected from real data.** We draw driving scene background assets in bird's eye view (BEV) in (a).

data synthesis, we propose SafeScale , a geometry-based generative driving simulation method. We employ geometry-based scene construction to explicitly combine assets in driving scenes, including traffic participant appearances, static backgrounds, and agent behavioral patterns. This enables centimeter-precise control over scene content, and on-demand batch generation of typical corner cases via nearly infinite combinations of assets sampled from large-scale asset libraries. Generative models are employed to synthesize photorealistic camera observation images that align with the constructed geometric scenes and simulated trajectories. By integrating **geometric** scene modeling with **generative** view rendering, our method provides a scalable and reliable solution for large-scale, high-quality driving data synthesis.

## 3.1 DRIVING SCENE DECOMPOSITION

Key elements in a driving scene that have significant influences on ego decision-making include the driving background, the appearance attributes of other traffic participants, and their behaviors. To enable modular and scalable generation of diverse driving data, we require large-scale, decoupled libraries of these three types of elements. At the same time, to achieve scalability of the pipeline, it is necessary to obtain these element assets at as low a cost as possible.

Therefore, we represent scene backgrounds and traffic participant appearances using colored 3D point clouds, which can be automatically and efficiently extracted from real driving data. We extract driving scene assets to form asset libraries from each existing real driving scene in the dataset. Specifically, LiDAR points from each frame of a real data clip are projected onto the camera plane to record their colors. Using object annotations, we separate points belonging to $N$ traffic participants from the scene background, resulting in $N + 1$ point cloud subsets. Point clouds are accumulated across frames: background points in the world coordinate system, and traffic participants along their trajectories. In this work, we extract the driving scene element asset library from the large-scale NAVSIM benchmark.

**Background library**. We remove all traffic participants from each scene based on annotations, extracting 1192 backgrounds from the NAVSIM benchmark. These driving backgrounds cover diverse road types, including urban arterials, intersections, and rural roads, as shown in Fig.2(a). Due to the

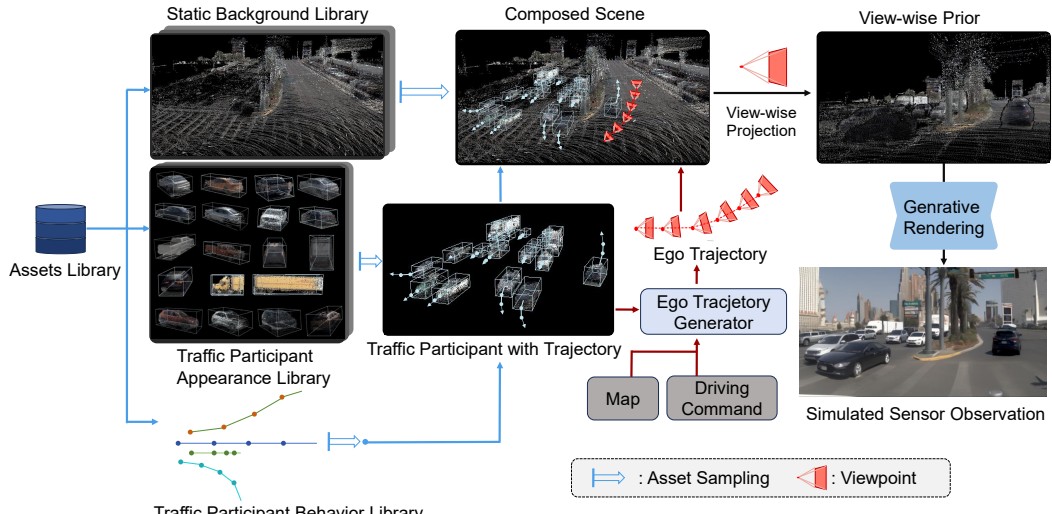

Figure 3: **Method pipeline of SafeScale.** We propose to geometrically represent and compose driving scenes to achieve precise scene content control. A generative model is utilized to synthesize camera views along the simulated ego trajectory in the geometrically composed novel scene.

extensive spatial coverage of the original data clips, most backgrounds exhibit large spatial scales and diverse road compositions.

**Traffic participant appearance library.** From each data clip, we extract colored point clouds of traffic participants, along with their semantic categories and 3D bounding box dimensions. This enables precise retrieval of traffic participant types when constructing new scenes. We collect 71.2K traffic participant appearances, each consisting of over 3K 3D points, including diverse categories such as sedans, trucks, vans, and semi-trailer trucks, as shown in Fig.2(b). Due to the limited observation angles towards traffic participants in the raw data, most traffic participant appearance observations we collected are partial, revealing the complete object appearance only from limited viewpoints. To characterize this, we compute and record the **object visible angle** of objects, which is defined as the direction angle of the visible surface in **the object's local coordinate system**, judged by the line of view from the ego position, as illustrated in the appendix, Fig.A.

**Traffic participant behavior library.** This library contains trajectories of dynamic participants, represented by the center positions and yaw angles of their 3D boxes across frames. Trajectories are categorized by mean speed, direction (straight, left-turning, right-turning), and speed profile (constant, accelerating, decelerating). We collected 68.8K trajectories from the NAVSIM training set.

Our extraction pipeline, relying solely on sensor data and 3D annotations, is highly scalable. The resulting large-scale library significantly enhances the diversity of generated scenes. Furthermore, the decoupled libraries allow us to conveniently and systematically supplement scarce assets to the asset library, such as rare road backgrounds or uncommon traffic participants.

## 3.2 GEOMETRIC NOVEL SCENE COMPOSITION

Based on the asset libraries extracted from real datasets, we can generate diverse novel driving scenes at scale by independently sampling assets for novel scene composition.

**Scene background selection.** We first select a static driving background from the library and specify the initial position of the ego vehicle. Benefiting from the generative model's ability to synthesize free-view observations, camera views can be simulated at arbitrary locations in driving scenes, without being restricted by the ego vehicle's original trajectory in raw data.

**Defining traffic participants.** We insert randomly or deliberately sampled traffic participants into the background and assign their trajectories. The traffic participants' appearances and behaviors

are independently sampled from the respective visual or behavioral asset libraries. By default, the initial positions of traffic participants are constrained within the drivable areas, typically near lane centers. Their trajectories are sampled to approximately follow nearby lane directions. For each inserted traffic participant, we calculate its object visible angle and sample an object appearance asset, i.e., a colored object point cloud, from the library with a close object visible angle. This sampling strategy ensures that the inserted traffic participant's appearance integrity is not affected by the partial point cloud assets. The sampled object appearance, i.e., colored 3D points normalized to the object coordinate system, is then inserted into the scene point cloud along its assigned trajectory in the novel scene per frame.

**Scene validity constraints.** During novel scene composition, we perform scene validity checks when inserting each traffic participant to ensure that the traffic participant behaves plausibly (e.g., roughly following lane directions, staying within drivable areas, and avoiding collisions with other participants), unless abnormal behavior is explicitly required. Participants are inserted sequentially. For each object, we check in the bird's eye view (BEV) that its bounding box lies within drivable areas and does not overlap with other objects. The same procedure guarantees that the ego vehicle's historical trajectory within the constructed scene is valid and non-abnormal by default.

We summarize the geometric novel scene composition pipeline in Fig.3. Implemented based on either vectorized or point cloud scene representations, the proposed geometric scene composition pipeline is computationally efficient, supports batched processing, and thus serves as a key component for scalable driving data synthesis.

Table 1: **Statistics on the failure cases of baseline DiffusionDrive planner on NAVSIM benchmark.**

| Failure Case Type | Number | Percent |
|---|---|---|
| **Collision** | 227 | 100% |
| Rear-End Collision | 177 | 78% |
| Other | 50 | 22% |
| **Near Collision** | 653 | 100% |
| Near Rear-End Collision | 551 | 84% |
| Other | 102 | 16% |
| **Drivable Area Deviation** | 493 | 100% |
| Lateral Drift | 436 | 88% |
| Other | 57 | 12% |

## 3.3 REQUIREMENT-DRIVEN SCENE COMPOSITION

As discussed above, SafeScale is designed to purposefully and scalably synthesize corner-case driving data. To generate driving scenes targeting weaknesses of end-to-end planners, we first analyze failure cases of the baseline DiffusionDrive(Liao et al., 2025) planner on the NAVSIM test set, as reported in Table 1. We report the failure cases where the predicted ego trajectory (i) collides with other road users, (ii) gets dangerously close to other road users, or (iii) leaves the drivable area.

**Scenarios with collision risk.** Among collision or near-collision (maintaining the current ego motion state would lead to a collision within several frames) failure cases, the majority involve rear-end collisions with a near-front vehicle (vehicle within 20m ahead and ±2m laterally of the ego vehicle). To improve planner performance under such driving scenarios, we leverage SafeScale to specifically synthesize cases with rear-end risk. We first compose a random novel scene with no front vehicle. Then, we sample a decelerating trajectory from the behavioral library and pair it with a random vehicle appearance. This traffic participant is inserted in front of and near the ego vehicle, thereby constructing a scenario with rear-end risk.

**Scenarios with driveable area aeviation risk.** We also observed that in failure cases where the baseline DiffusionDrive model predicts a trajectory extending beyond the drivable area, the ego vehicle typically collides with the left or right lane boundaries while moving forward. This occurs because the planner fails to keep the ego vehicle near the lane center or to predict corrective trajectories after deviation. We conjecture that this tendency arises because the NAVSIM dataset, which comprises driving data obtained from human drivers exhibiting safe and conservative behavior, largely

lacks trajectories that once deviate from the lane center. This prevents the imitative planner from learning to avoid or recover from lane deviations. To address the shortage of corresponding real-world data, we employ SafeScale to specifically generate driving scenarios with abnormal initial ego-vehicle states. We first randomly compose new driving scenarios, and then set the historical trajectory and current position of the ego vehicle close to the road or lane boundaries.

Following the same approach, we can also generate other corner cases in bulk, such as scenarios with wrong-way vehicles or aggressive lane-cutting vehicles, as shown in Fig.1. However, since such scenarios are originally absent in the NAVSIM test set, we do not include these corner cases in the scope of synthetic data scaling for improving end-to-end planning in this paper.

### 3.4 Camera Observation Simulation

After constructing the novel driving scenarios, we synthesize the camera observations from the ego vehicle's viewpoint for each frame along its trajectory using a generative model. Following the design of the generative novel view synthesis (NVS) method FreeVS(Wang et al., 2025), we project the point cloud representation of the novel driving scenario into an affine viewpoint to generate the pseudo-image representation of the 3D scene prior, and then use the generative model to produce the camera observations based on the pseudo-image prior. The rendering process based on the geometric prior strictly ensures consistency between the simulated camera observations and the specified ego trajectory, achieving zero-execution-error trajectory simulation. Introducing generative rendering also enables us to generate camera views based on low-cost point cloud scene representations, without relying on high-cost, fine-grained 3D assets, thereby ensuring the scalability of the proposed pipeline. To accommodate the relatively sparse LiDAR observations and longer video sequence lengths in the NAVSIM dataset, we improved the point cloud projection operation in the FreeVS method, ensuring that 3D content at different depths and between foreground and background is less likely to be confused in the perspective view.

### 3.5 Trajectory Label Generation

In this work, we train the end-to-end planner following the imitation learning paradigm. For the synthesized new scenarios, we use the PDM planner(Dauner et al., 2023), the winning solution in the 2023 nuPlan planning competition, to search for a safe driving trajectory within the novel driving scenario as the learning target for the end-to-end planner. Specifically, we modified the PDM planner to ensure that it maintains a sufficient safety distance from the leading vehicle in rear-end risk scenarios, and actively steers the ego vehicle back toward the lane center when it deviates laterally. More details on trajectory label generation are illustrated in the appendix.

## 4 Experiments

In this section, we first introduce our experimental setup, including datasets and method implementation. Then we provide the main and ablation experiment results.

**Dataset.** We conduct experiments on driving data synthesis and end-to-end planner training using the NAVSIM(Dauner et al., 2024) benchmark. The NAVSIM benchmark is built on the OpenScene dataset(Contributors, 2023), which contains 120 hours of driving logs selected from the nuPlan dataset(Caesar et al., 2021). The OpenScene dataset is constructed by resampling the nuPlan dataset and emphasizes the planner's handling of non-trivial driving scenarios beyond simple straight-line driving. The NAVSIM dataset defines the Predictive Driver Model Score (PDMS) to evaluate planner performance. PDMS is computed based on a combination of elements, among which the no at-fault collision (NC) and time-to-collision (TTC) metrics assess the planner's collision risk with other vehicles, while drivable area compliance (DAC) evaluates whether the planned trajectory adheres to drivable areas. In our experiments, we include the number of failure cases of the end-to-end planner on these three metrics as part of our observations.

**Implementation of SafeScale.** We decompose all real driving scenarios in the NAVSIM Training Set into a total of 1192 static scene background assets, 71.2K traffic participant appearance assets, and 68.8K traffic participant trajectory assets. Each traffic participant appearance asset is extracted from at least 14 consecutive frames and is composed of more than 3K LiDAR points.

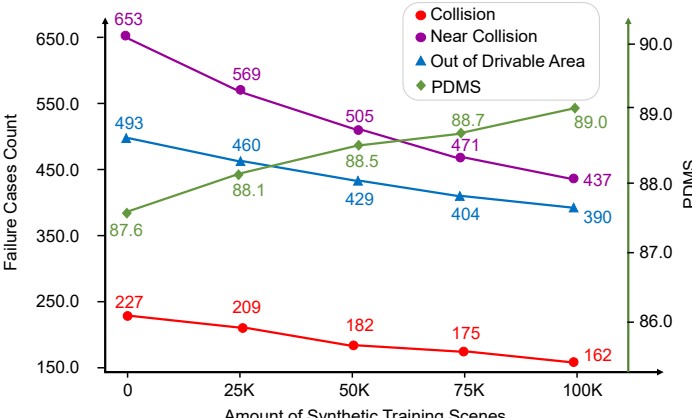

Figure 4: **The impact of the amount of synthesized data.** We incorporate the synthesized random scenes or targeted corner scenes into the real training data as additional training samples. As the amount of synthesized training data increases, the number of failure cases of the end-to-end planner continuously decreases, revealing a clear data scaling effect. Best viewed in color.

Trajectories of static objects are excluded. After projecting the point cloud representation of the composed novel scenarios into the target viewpoint to form a pseudo-image, we use a Stable Video Diffusion model(Blattmann et al., 2023) to render camera observations based on the pseudo-image prior. We implement the generative model following the generative novel view synthesis method FreeVS(Wang et al., 2025). The training schedule is extended to 50K iterations.

**Implementation of end-to-end planner.** We select the state-of-the-art (SOTA) DiffusionDrive(Liao et al., 2025) planner as the baseline planner. We report the performance of the baseline Diffusion-Drive planner trained on the NAVSIM training set, as well as the performance of the DiffusionDrive planner trained on both the NAVSIM training set and the synthesized data, evaluated on real driving data from the NAVSIM test set.

## 4.1 EXPLORING SYNTHETIC DATA SCALING EFFECT

In this section, we investigate the impact of scaling up synthetic training data on the performance of the end-to-end planner. We first generate scaled driving data using randomly composed scenarios, and then supplement the synthetic data with targeted rear-end risk scenes and drivable area deviation risk scenes. After a simple grid search, we mix these three types of synthetic data in a 4:3:3 ratio. Specifically, for up to 100K synthesized driving scenarios, 40K of them are randomly composed new driving scenarios, 30K of them are synthetic rear-end risk scenarios, and 30K of them are synthetic scenarios with drivable area deviation risk. During the training of the end-to-end planner, the synthetic data are added to the initial training set consisting of 103288 real driving scenarios from the NAVSIM training set.

Fig.4 illustrates the impact of scaling up the synthetic training data amount on the end-to-end planner. We observe that the introduction of any amount of synthetic training data provides positive gains to end-to-end planner performance on the real-world test set, indicating that the sim-to-real gap of SafeScale -synthesized data is small and does not cause a negative impact. More importantly, as we scale up the amount of synthetic training data, the number of collision and drivable area deviation failure cases of the end-to-end planner steadily decreases, while its PDMS performance continues to improve. This demonstrates that training the end-to-end planner with large-scale synthetic data indeed follows a synthetic data scaling law, with the model continually benefiting from the increasing amount of synthetic driving data.

## 4.2 ABLATION STUDIES

**Scaling effect of synthetic corner case data.** In Fig.5(a) and Fig.5(b), we show the impact of adding only the synthetic rear-end risk scenes or scenarios with drivable area deviation risk to the training set. The experiments indicate that the addition and scaling up of rear-end risk scenes consistently

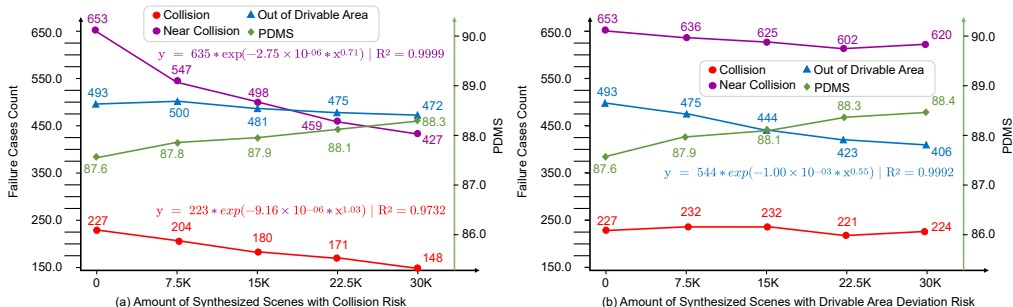

Figure 5: **The impact of the amount of synthesized corner case data.** In the figure, we annotate the fitted relationship between synthetic data volume and the number of failure cases using the same color as the corresponding line.

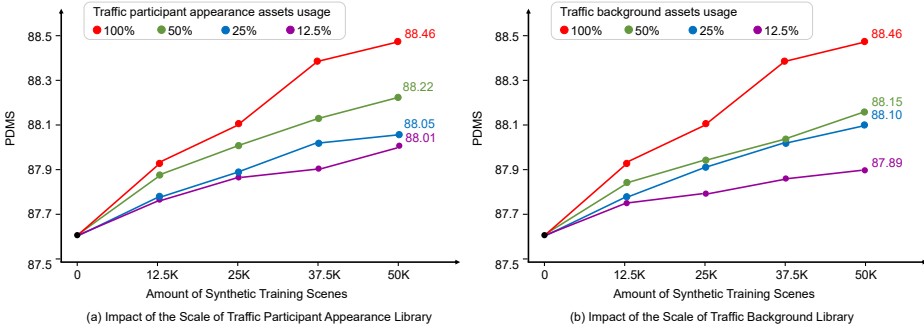

Figure 6: **The impact of the scale of driving scene asset libraries.** We investigate how the scale of a library (i.e., the number of assets) used to generate synthetic training data affects an end-to-end planner's performance, while the total volume of synthetic data is held constant.

reduce the number of collision and near-collision fail cases for the end-to-end planner, but have little effect on the number of drivable area deviation fail cases. Conversely, the addition of drivable area deviation risk scenes reduces the number of road departure failure cases, while showing a negligible impact on the number of collision failure cases. This demonstrates that targeted synthesis of specific corner cases as training data can selectively strengthen the planner's performance in the corresponding corner scenarios. Moreover, comparing with the results in Fig.4, the optimization effects of SafeScale-synthesized data on different types of corner cases are independent and additive, allowing the safety of end-to-end planning to be improved in a modular fashion by composing targeted synthetic data.

**Impact of the scale of the driving scene element libraries**. The scale and diversity of the driving scene element libraries directly affect the diversity of novel scenarios that SafeScale can construct. In Fig.6, we compare the effect on the end-to-end planner of synthetic training data generated using libraries of different scales while keeping the total amount of synthetic data unchanged. Experiments demonstrate that increasing the amount of available driving scene element assets significantly improves the end-to-end planner's benefit from synthetic data.

## 5 CONCLUSION

In this paper, we focus on the scalable synthesis of driving data, especially corner data, for enhancing end-to-end planning. We propose SafeScale, a geometry-based generative driving simulation method. We decompose real driving data datasets into libraries of driving scene components, and explicitly compose driving scene components to construct random or targeted novel driving scenes. Generative model is utilized to simulate camera observations in the composed novel scenes. Experiments show that scaled training data synthesized by SafeScale can significantly boost the performance of the SOTA end-to-end driving planner, revealing a clear synthetic data scaling effect.

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

# A APPENDIX

Table A: **Comparisons with recent works on the NAVSIM test set.** *: We report the performance of the baseline DiffusionDrive model based on its open-sourced official implementation. †: From Li et al. (2024b).

| Method | Input | NC↑ | DAC↑ | EP↑ | TTC↑ | Comf↑ | PDMS↑ |
|---|---|---|---|---|---|---|---|
| Human† | - | 100 | 100 | 87.5 | 100.0 | 99.9 | 94.8 |
| UniAD(Hu et al., 2023b) | C | 97.8 | 91.9 | 78.8 | 92.9 | 100.0 | 83.4 |
| TransFuser(Chitta et al., 2022) | C&L | 97.7 | 92.8 | 79.2 | 92.8 | 100.0 | 84.0 |
| Hydra-MDP(Li et al., 2024c) | C&L | 98.3 | 96.0 | 78.7 | 94.6 | 100.0 | 86.5 |
| DiffusionDrive(Liao et al., 2025) | C&L | 98.2 | 96.2 | 82.2 | 94.7 | 100.0 | 88.1 |
| DiffusionDrive* | C&L | 98.2 | 95.9 | 81.8 | 94.6 | 100.0 | 87.6 |
| DiffusionDrive*+Ours | C&L | 98.8 | 96.8 | 82.0 | 96.4 | 100.0 | 89.0 |

## A.1 STATE-OF-THE-ART COMPARISON

We provide a state-of-the-art comparison between previous end-to-end planners and the Diffusion-Drive planner trained with or without data synthesized with SafeScalein Table A. We train the baseline Diffusiondrive planner following its official implementation without any modification on the NAVSIM benchmark and report its performance, marked by * in Table A.

## A.2 ILLUSTRATION OF OBJECT VISIBLE ANGLE

We illustrate the definition of the object visible angle of traffic participants in Fig A. When extracting object appearance assets from real data clips, we calculate the mean object visible angle label of an object by taking the weighted average of its object visible angle in each frame, weighted by the number of 3D points belonging to the object in each frame. When composing novel scenes, for each traffic participant, we define its object visible angle as its visible angle at the middle frame of its total lifespan.

## A.3 DETAILS IN PSEUDO-IMAGE GENERATION.

The original implementation of FreeVS was designed for the Waymo Open Dataset (WOD), where the dense and uniform LiDAR observations allowed FreeVS to generate high-quality 3D scene priors with only a small number of frames (5 frames) of temporal LiDAR fusion. In contrast, the LiDAR observations provided in the NAVSIM benchmark are sparser and more irregular. Therefore, we accumulated LiDAR point clouds over a longer temporal sequence (60 frames). Additionally, when projecting the 3D point clouds of a new driving scenario into the camera view, we projected the 3D point cloud from near to far based on depth intervals. Within each depth interval, we first

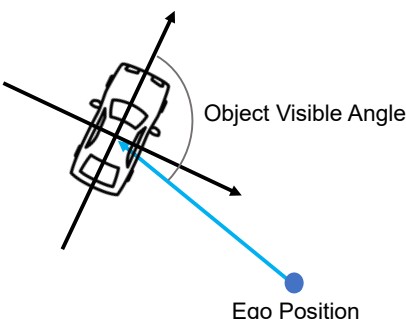

Figure A: **Definition of object visible angle.**

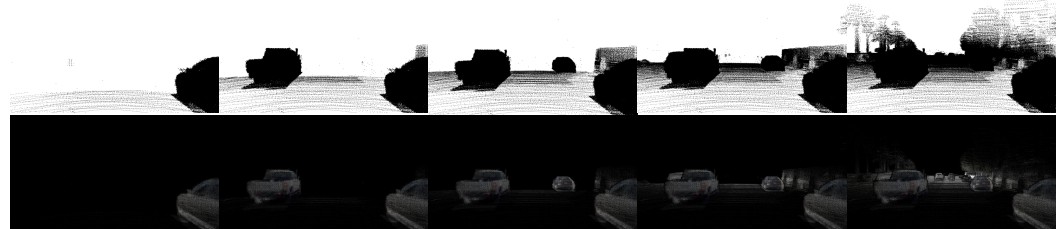

Figure B: **Illustration of the generation process of pseudo-images with occlusion-aware point cloud projection.** From left to right, we present the visualization results of projecting point clouds in depth order and foreground-background order while simultaneously updating the projection occupancy mask.

sequentially project each foreground object in depth order, and then project the point cloud of the static background. During each projection operation, we marked the neighborhood (radius 4 pixels) of the 2D projection points of each 3D point cloud in the camera view as occupied, and avoided projecting 3D points into the already occupied areas when projecting content. This occlusion-aware point cloud projection approach prevented overlapping and confusion caused by projecting sparse point clouds from different depths into the camera view.

## A.4 DETAILS IN TRAJECTORY LABEL GENERATION.

As discussed in (Dauner et al., 2023), the rule-based PDM-closed planner demonstrates strong and stable performance in closed-loop planning. The NAVSIM Benchmark also uses the closed-loop reasoning results of the PDM-closed planner as a baseline to evaluate the performance of open-loop planning methods.

PDM-closed planner generates a set of trajectory candidates by varying speeds and lateral offsets relative to the nearest lane line. It then simulates and scores each candidate in a closed-loop setting, evaluating them based on safety (collision avoidance), progress, driving area compliance, and comfort. The highest-scoring candidate is preserved. However, the inferred trajectory is essentially determined by the ego vehicle's current position and the direction of the nearest lane line. The resulting trajectory is approximately parallel to the nearest lane line and does not gradually converge toward the lane center, as illustrated in Fig.C.

As described in Sec.3.3, we aim to enhance the capability of end-to-end planners in lane-following and recentering when the ego vehicle deviates, by synthetic training scenarios with drivable area deviation risks. To achieve this, we modify the trajectory reasoning strategy of the PDM-closed planner so that the planned trajectory gradually approaches the reference lane center over future frames. This is implemented by taking an interpolation between the originally inferred trajectory

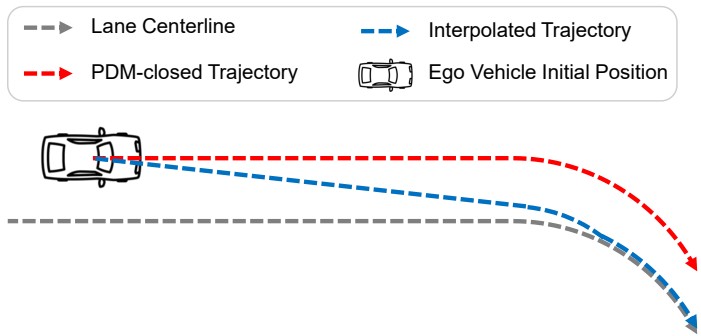

Figure C: **Illustration of the lane-following-enhanced trajectory label.**

and the reference lane center, as shown in Fig.C. Meanwhile, when generating trajectory labels, we gradually increase the safe distance value for maintaining distance to the leading vehicle ([4m, 8m, 12m, 20m]) and decrease the maximum speed ([1.0, 0.6, 0.4, 0.2, 0.05] times the road speed limit) in the PDM-closed planner, until a trajectory label that achieves full scores in both collision and near-collision metrics is obtained. Through these designs, we reinforce the reference trajectory's tendency to maintain a safe distance from the preceding vehicle and adhere to the lane center, thereby enhancing the beneficial effect of synthesized corner cases on the end-to-end planner.

## A.5   LARGE LANGUAGE MODEL USAGE.

Large Language Models (LLMs) are used for polishing writing in this manuscript.

