# OpenReview forum: "Scaling Autonomous Driving Safety with Synthetic Data"
_ICLR.cc/2026/Conference — ICLR 2026 Conference Withdrawn Submission_

### Official Review · Reviewer_KWWg · 2025-10-29

**Soundness:** 2
**Presentation:** 3
**Contribution:** 2
**Rating:** 4
**Confidence:** 4

**Summary:**

This paper presents a synthetic data augmentation method for improving the performance of end-to-end autonomous driving planning systems. The authors propose a pipeline that decomposes real-world driving scenarios into foreground objects and background environments to create a library of assets. These assets are then used to generate novel driving scenarios, which are synthesized for training an autonomous driving planner. The authors demonstrate that incorporating synthetic data into the training set improves open-loop planning performance, as depicted by reduced collisions, off-road events, and improved open-loop metric PDM Score.

**Strengths:**

1. The authors introduce a systematic approach to generate diverse driving scenarios by leveraging synthetic data. The quality of rendered images and the diversity of synthesized scenarios are commendable.

2. The experiments show that adding synthesized data to the real-world dataset results in measurable improvements in the planner's performance, reducing collisions and off-road events. This addresses a critical gap in current data-driven planning methods.

3. The proposed method utilizes a novel asset library that separates foreground objects and background environments, enabling scalable scenario generation. This approach has the potential to significantly enhance the diversity of corner cases in autonomous driving without the need for additional real-world data collection.

**Weaknesses:**

1. One of the major weaknesses of the paper is that the performance gains from incorporating synthetic data are relatively modest. Doubling the dataset with 100K synthesized frames results in only a 1.4% increase in the PDMS metric. This suggests that the synthetic data does not have a major impact on the planner's performance, and the original dataset still dominates.

2. While the method is demonstrated on DiffusionDrive, it is not clear whether this approach can be applied to other end-to-end planning frameworks. The authors should explore the generalizability of this approach to other architectures (e.g. Transfuser) to better understand its broader applicability.

3. The rule-based PDM-Planner is used to predict ego-vehicle trajectories, which may limit the diversity and complexity of the synthesized scenarios. If the planner itself is overly conservative or lacks diversity in trajectory generation, the synthetic data may not cover a wide range of driving situations, thus reducing the overall performance improvements.

4. The evaluation focuses primarily on open-loop planning metrics, such as collision rates and off-road events. However, driving performance in closed-loop settings (e.g. CARLA or Bench2Drive), should also be considered to evaluate the system’s ability to react to dynamic and unforeseen situations.

**Questions:**

Please refer to the weaknesses section. I do not have other questions.

---

### Official Review · Reviewer_Zjb3 · 2025-10-30

**Soundness:** 2
**Presentation:** 3
**Contribution:** 3
**Rating:** 4
**Confidence:** 4

**Summary:**

The paper introduces SafeScale, a generative simulation framework designed to synthesize large-scale, diverse, and precisely controllable driving corner cases for improving the robustness of planners. SafeScale decomposes real-world driving data into reusable visual and behavioral assets, and recombines them to generate novel 3D driving scenes. A generative rendering model is employed to produce photorealistic camera observations along simulated ego-vehicle trajectories, maintaining scene consistency and physical plausibility. Experiments on the NAVSIM benchmark demonstrate that supplementing the planner yields substantial improvements.

**Strengths:**

1. SafeScale is a geometry-based generative simulator that enables controllable and scalable synthesis of rare or safety-critical driving scenarios, addressing a key limitation in existing autonomous driving datasets.

2. By disentangling real-world driving data into static, visual, and behavioral components, the framework supports fine-grained customization and diverse recombination for generating realistic 3D driving scenes.

3. The approach is compatible with existing end-to-end planners and could be easily integrated into broader autonomous driving pipelines for simulation, testing, and training purposes.

**Weaknesses:**

1. The paper does not clarify how static or slow-moving vehicles are treated during background decomposition. Removing them might cause structural holes or visual discontinuities in the reconstructed static backgrounds.

2. Although SafeScale ensures consistent viewpoints, it remains unclear how it handles distance-dependent visual degradation, such as when low-resolution point assets are brought close to the camera, potentially leading to perceptual artifacts.

3. The method lacks details on how traffic and collision constraints are encoded or verified to ensure physically and behaviorally valid trajectories across large-scale synthetic datasets: a major engineering challenge.

4. The paper omits key implementation aspects of the generative rendering model, including its training data and architectural design, making it difficult to assess photorealism and generalization quality.

5. No visual comparisons are showing which baseline failures SafeScale mitigates, nor ablation experiments to isolate the effects of individual components.

6. Typos: Figure 3 "Genrative".

7. Missing related work: Nexus (ICCV 2025).

**Questions:**

Please refer to the weaknesses above.

---

### Official Review · Reviewer_1fwp · 2025-11-01

**Soundness:** 2
**Presentation:** 3
**Contribution:** 3
**Rating:** 4
**Confidence:** 3

**Summary:**

This paper presents SafeScale, a novel and scalable framework for generating synthetic driving data to improve the safety and robustness of autonomous driving planners. The core problem addressed is the difficulty of collecting real-world data for rare and dangerous "corner cases" that often cause modern data-driven planners to fail.

**Strengths:**

Strengths
1.	The paper tackles a critical and high-impact problem in autonomous driving: improving safety by addressing corner cases. The primary contribution—a clear, empirical demonstration that scaling targeted synthetic data can directly and significantly improve a planner's performance in the real world.
2.	The SafeScale framework is well-designed and technically sound. The idea of decomposing real scenes into modular asset libraries (backgrounds, appearances, behaviors) is a powerful concept that enables both scalability and fine-grained control.
3.	The experimental evaluation is exemplary. The main result, presented in Figure 4, provides convincing evidence of a "synthetic data scaling law," which is the paper's central claim. The ablation studies are insightful and strongly support

**Weaknesses:**

Weaknesses:

1.	The contribution can be characterized as a sophisticated and highly successful data engineering framework. Its primary novelty lies in the clever integration of existing components and the powerful empirical demonstration of the scaling law, rather than in a fundamental algorithmic advance. The method for generating corner cases is 'reactive'—it relies on first analyzing the specific failure modes of a baseline planner (DiffusionDrive) on a specific benchmark (NAVSIM). A more scientifically profound direction, which the paper does not explore, is how to proactively and universally model and generate corner cases. For instance, could one develop a general model that learns the underlying distribution of safe driving data and then synthesizes challenging out-of-distribution scenarios in a principled way, without being tied to the failures of one specific planner? This would elevate the contribution from a highly effective, bespoke data augmentation solution to a more fundamental model of driving risk.
2.	The traffic participant behavior library is built from trajectories observed in the NAVSIM dataset. While the framework can create novel scenarios by placing these behaviors in new contexts, it is fundamentally limited by the vocabulary of behaviors present in the source data. The paper would be strengthened by a discussion of this limitation. Can this method generate truly novel, out-of-distribution behaviors, or is it primarily a powerful recombination engine?
3.	The paper states that the asset extraction pipeline is highly scalable as it relies on sensor data and 3D annotations. However, the acquisition of clean, large-scale 3D annotations is a known bottleneck and a significant cost factor for the entire industry. A brief discussion on the sensitivity of the SafeScale pipeline to the quality and scale of these initial annotations would be welcome. For instance, how do sparse or noisy annotations affect the quality of the generated assets and the final planner performance?
4.	The use of a generative model for view synthesis is a key strength, but these models are not perfect. They can introduce subtle artifacts, temporal inconsistencies, or a lack of physical realism (e.g., incorrect shadows, reflections). The paper does not discuss the potential sim-to-real gap of this rendering stage. While the end-to-end results prove the data is highly effective, a qualitative analysis or discussion of the generative renderer's failure modes would add nuance and provide a more complete picture

**Questions:**

na

---

### Official Review · Reviewer_VHdN · 2025-11-02

**Soundness:** 2
**Presentation:** 2
**Contribution:** 2
**Rating:** 2
**Confidence:** 4

**Summary:**

This paper presents SafeScale, a geometry-based pipeline for synthesizing autonomous-driving training data at scale. The method decomposes real scenes into modular asset libraries and composes new scenes by procedurally sampling and combining these assets under plausibility constraints, then renders camera observations along ego trajectories with a generative novel-view model. Using NAVSIM, the authors curate random scenes and targeted corner-case scenarios (rear-end risk; drivable-area deviation) and train an end-to-end planner (DiffusionDrive) on real + synthetic data. They report a data-scaling effect, with increasing synthetic volume improving NAVSIM test performance and reducing failure cases.

**Strengths:**

- Scaling up synthetic data on NAVSIM improves metrics.** The study demonstrates a **monotonic improvement** trend as synthetic data grows (random + targeted), with corresponding reductions in failure counts and PDMS gains; targeted corner-case data improves the matching failure modes, suggesting **selective, additive** benefits.

- **Quantified, task-relevant evaluation.** The paper ties results to NAVSIM’s PDMS (especially NC/DAC components) and provides per-failure-type analyses and ratios (e.g., rear-end prevalence), helping interpret where synthetic data is most beneficial.

**Weaknesses:**

- **Vague core contribution in scene composition.** The core novelty primarily lies in asset recomposition with generative *rendering* of views. As written, the title and framing risk over-stating a generative scenario-creation capability relative to the underlying rule-/asset-based composition; moreover, the paper acknowledges that important long-tailed scenarios are absent from NAVSIM, which limits evidential support for claims about covering such tails. Meanwhile, prior works like [1, 2] have already utilized
- **Limited baseline coverage.** The principal planner studied is **DiffusionDrive**. While the appendix lists prior methods, the main scaling analyses largely hinge on a single SOTA baseline, which weakens claims of a **general scaling law** for synthetic data across model families.
- **Counter-trend in near-collision at high synthetic counts.** In Fig. 5 (right), **near-collision** failure counts **increase** between ~22.5K and 30K synthetic samples; the text does not analyze this **regression** or provide diagnostics for why scaling occasionally hurts specific submetrics.
- **Potential label-generation bias.** Ground-truth trajectories for synthetic scenes are produced by a **modified PDM-closed** planner with strengthened lane-centering and distance-keeping heuristics, which could imprint planner-specific priors and partially account for metric improvements (especially DAC/NC) independent of true distributional realism. More ablations and experiment results would be requierd to demonstrate the effectiveness of scalable synthetic data.

> [1] Wu, Yanhao, et al. "Generating Multimodal Driving Scenes via Next-Scene Prediction." CVPR 2025.
>
> [2] Wang, Jiahao, et al. "Drive&Gen: Co-Evaluating End-to-End Driving and Video Generation Models.", IROS 2025.

**Questions:**

- Could you precisely distinguish the components that rely on a generative model and the ones that rely on a procedural generation process? For instance, are behaviors synthesized by a learned model or solely sampled from a trajectory library with constraints? What prevents **combinatorial asset recomposition** from saturating and failing to yield **novel, rare** interactions beyond those already in the library?

- In line 330-331, you note that some corner cases (e.g., wrong-way, aggressive cut-ins) are **absent from the NAVSIM test set**, hence excluded from scaling experiments. Can you (i) quantify coverage gaps and (ii) evaluate on an external benchmark or bespoke test suite containing these long tails to substantiate the “safety scaling” claim beyond NAVSIM?

- Do scaling trends hold for **different E2E planners** (e.g., transformer-fusion, model-based IL)? Please report the results of SafeScale on additional strong baselines (e.g. UniAD, VAD, VAD-v2, etc.) from NAVSIM to ensure the observed **scaling law** is not model-specific.
- What accounts for the **increase** in near-collisions at higher synthetic volumes? Did you examine sampling ratios (random vs. targeted sets), curriculum effects, or covariate shift in asset distributions (e.g., more partial-view appearance assets at certain visible angles) that might degrade TTC? Please provide ablations/diagnostics (e.g., per-scenario-type breakdowns) around 22.5K–30K.
- Since labels are produced by a **modified PDM-closed** planner enforcing lane-centering and conservative headway, how do you distinguish **true sim-to-real benefit** from simply learning the teacher’s bias (which directly optimizes PDMS submetrics)? Could you compare against the unmodified PDM label to assess the applicability of SafeScale?
- More recent benchmarks attempt to use more closed-loop-aligned evaluation [1, 2]. It would be interesting to see if SafeScale performs well on these new benchmarks.

> [1] Zhou, Hongyu, et al. "Hugsim: A real-time, photo-realistic and closed-loop simulator for autonomous driving." *arXiv preprint arXiv:2412.01718* (2024).
>
> [2] Cao, Wei, et al. "Pseudo-simulation for autonomous driving." CoRL 2025.

---

### Note · Authors · 2025-11-14

I have read and agree with the venue's withdrawal policy on behalf of myself and my co-authors.